# Futility in TAVI: A scoping review of definitions, predictive criteria, and medical predictive models

**Charlie Ferry**[ID][1]*, **Jade Fiery-Fraillon**[2], **Mario Togni**[1], **Stephane Cook**[1]

**1** Department of Cardiology, University & Hospital Fribourg, Fribourg, Switzerland, **2** Institute of Higher Education and Research in Healthcare–IUFRS, Lausanne, Switzerland

* charlie.ferry@unifr.ch

## Abstract

### Background

Transcatheter Aortic Valve Implantation (TAVI) procedures are rapidly expanding, necessitating a more extensive stratification of patients with aortic stenosis. Especially in the high-risk group, some patients fail to derive optimal or any benefits from TAVI, leading to the risk of futile interventions. Despite consensus among several experts regarding the importance of recognizing and anticipating such interventions, the definition, and predictive criteria for futility in TAVI remain ambiguous.

### Aim

The purpose of this study is to explore the literature addressing the definition, predictive criteria, and medical predictive models for futility in cases of TAVI.

### Design

A scoping review was conducted by two researchers and reported in accordance with the PRISMA-ScR guidelines.

### Eligibility criteria

Studies addressing futility in TAVI, including definitions, predictive variables, and models, were included without restrictions on study design but were excluded study only on surgical valve replacement, valve in valve or aortic stenosis causes by other pathology than calcification.

### Information sources

We identified 129 studies from five key sources: CINAHL, PUBMED, the Cochrane Library, ClinicalTrials.gov, and EMBASE. The literature search was conducted in two rounds—first in February 2024 and again in October 2024—using no restrictions on the year of publication or the language of the studies. Additional references were included through cross-referencing.

modifications made. The search equations used to access the relevant articles are provided in Appendices 1, 2, 3, and 5.

**Funding:** The author(s) received no specific funding for this work.

**Competing interests:** The authors have declared that no competing interests exist.

**Abbreviations:** ACC, American College of Cardiology; AS, Aortic Stenosis; AUC, Area Under the Curve; CKD, Chronic Kidney Disease; CPI, Cardiac Power Index; COPD, Chronic Obstructive Pulmonary Disease; EFT, Essential Frailty Toolset; ESC, European Society of Cardiology; FTS, Futile TAVI Simple score; HG AS, High Gradient Aortic Stenosis; HF, Heart Failure; HR, Hazard Ratio; KCCQ, Kansas City Cardiomyopathy Questionnaire; LF LG, Low Flow Low Gradient; ML, Machine Learning; MR, Mitral Regurgitation; NYHA, New York Heart Association scale; PASP, Pulmonary Artery Systolic Pressure; pEF, preserved Ejection Fraction; pLFLG-AS, paradoxical Low-Flow, Low-Gradient aortic stenosis; PR, Pulmonary valve Regurgitation; PRISMA, Preferred Reporting Items for Systematic Reviews and Meta-Analyses; PVS, Plasma Volume Status; SAVR, Surgical Aortic Valve Replacement; SDM, Shared Decision Making; STS, Society of Thoracic Surgeons; TAVI, Transcatheter Aortic Valve Intervention; Zva, Valvulo arterial impedance.

## Results

The definition of futility is not unanimous, although most researchers agreed on 1-year survival as a cutoff. The majority of studies focused on single variables that can predict 1-year survival, employing either prospective or retrospective designs. Frailty was the major concept studied. Numerous predictive models have been identified, but no consensus was found.

## Conclusion

Futility concepts generate interest in the TAVI procedure. In this review, numerous articles state that 1-year mortality serves as a cutoff to define futile procedures. Some variables, cardiac or otherwise, are independent predictors of 1-year mortality. Medical predictive models showed moderate sensitivity and specificity, except for machine learning, which shows promise for the future. However, few articles delve deeply into non-quantifiable parameters such as patient goals and objectives or ethical questions. More studies should focus on these parameters.

## Introduction

Aortic stenosis (AS) is prevalent in individuals over 75, with around 5% affected [1], and is responsible for about 65% of valvular disease deaths [2]. Its incidence rises with age, often diagnosed late when symptoms like dyspnea, syncope, and retrosternal pain appear [2]. Treatment options for AS include Surgical Aortic Valve Replacement (SAVR), transcatheter aortic valve implementation (TAVI), or conservative medical approaches. TAVI, now offered to medium surgical risk patients, has seen increased use [3]. Guidelines recommend TAVI for patients over 75 or at high risk, but methods for patient stratification, particularly in evaluating the risk of futility, lack evidence [3]. While TAVI benefits are evident in survival, symptoms, and quality of life [4–6], concerns arise as around 20% of medium and high-risk patients may not survive the first year post-TAVI [7,8]. The term 'futility' in TAVI lacks consensus in definition and prediction criteria [9–11], and tools for pre-TAVI stratification are lacking [3]. A clear definition is essential as it influences patient access to TAVI, with some being denied due to perceived high risk of futile intervention [12], while others may not benefit significantly from the procedure.

This study aims to explore literature on futility in TAVI, focusing on understanding the criteria, both objective and subjective, defining futility, identifying predictive criteria indicating a TAVI procedure at risk of futility, examining non-predictive factors associated with futility in TAVI, and assessing available tools for stratifying TAVIs at risk of futility.

## Methods

### Ethics considerations

This research does not contain any human subjects.

**Protocol.** A scoping review was conducted following the checklist outlined in the PRISMA for Scoping Reviews (PRISMA-ScR) [13] (S8 Table). Several researchers debated the key concepts to identify research questions and search equations. These equations were employed to identify and choose studies that meet the eligibility criteria. Two researchers,

CF-JFF, independently screened the studies. The selected studies were then collated and sorted based on their relevance. A third step involved data collection and summarizing the results.

**Eligibility criteria.** The eligibility criteria were defined and used by FC and CFF as summarized in "Table 1".

**Information sources.** To retrieve potential articles and documentation related to our research topic, the libraries of PubMed, Cinahl, Embase, Cochrane Library, and ClinicalTrials. gov were consulted. All articles were exported into reference management software (Zotero) and duplicates were removed.

**Search.** The search strategy, developed with the assistance of the University Hospital Fribourg librarian, utilized key concepts (S1 and S2 Tables) to formulate search equations (see S3 Table). Additional articles were included through cross-referencing to supplement the initial database. At any point during the search, if an article appeared interesting to either of the two researchers, it was added to the screening process. The search strategy underwent scrutiny by three researchers following the Peer Review of Electronic Search Strategies checklist [14]. All search results were independently reviewed by two researchers. Language filters were not applied, as modern technology, including AI, enables reliable translation. Furthermore, no date restrictions were imposed, considering the recent emergence of the TAVI procedure, with timeless themes such as ethical considerations. However, studies were stratified based on their publication dates during results extraction (S6 Table), with particular attention to technological advancements.

**Selection of sources.** Two researchers screened the search results, initially based on the title, followed by the abstract, and finally, the full text (S4 and S5 Tables). The selection process was discussed and debated at each stage. In cases of disagreement or uncertainty, a third independent researcher was consulted to establish a consensus.

**Data charting process.** Two researchers collaboratively created a data mapping form to identify extraction variables. Each researcher independently conducted the data mapping, exchanged conclusions, and regularly adjusted the data mapping form in an iterative process.

**Data items.** The results were categorized based on several factors (S6 Table), including whether the article provided a definition of futility in the context of TAVI, proposed objective and subjective criteria for defining futility, described predictive criteria for futility, and utilized tools for stratifying pre-intervention futility.

**Results synthesis.** Following the extraction of key variables from the articles, we organized and categorized them in S7 Table for classification purposes.

**Table 1. Eligibility criteria.**

| Inclusion criteria | Exclusion criteria |
|---|---|
| **Population** | |
| **Patients with AS (with and without symptoms)** <br> **Patients with moderate to high surgical risk during TAVI** <br> **Frail Patients** | |
| **Concept** | Concept |
| **Studies reporting on futility in TAVI** <br> **Studies on Shared Decision Making in TAVI** <br> **Studies on stratification tools for TAVI** | Studies on Surgery Aortic Valve Replacement only <br> Studies on valve-in-valve procedures |
| **Design** | Context |
| **Any study design including expert opinion, cardiac clinic** <br> **protocol and position papers** | Aortic stenosis causes by other pathology than calcification <br> I.E post-endocarditis scarring <br> Rheumatic aortic stenosis |

# Results

## Source selection process

The search equations yielded 239 articles, with an additional 23 articles identified through cross-referencing. After removing 60 duplicates, screening based on title excluded 58 articles, primarily unrelated to TAVI or not addressing the question of futility. Subsequently, 36 articles were excluded based on abstract content, mainly focusing on futility criteria during the procedure rather than before. Following a full-text review of 108 articles, 13 were excluded due to unclear articulation of the futility concept. Ultimately, 95 articles were included. In October 2024, second research was performed on PubMed with a wider equation (S3 and S5–S7 Tables) to integrated newly published articles. A total of 29 new articles was fully read and 15 were included.

## Definition

Defining futility in TAVI is critical for patient outcomes and healthcare resource allocation. Arnold et al. describe futility as a failure to achieve the desired improvement in patient health or quality of life [15]. The American College of Cardiology (ACC) specifies TAVI as futile for patients with less than one year of life expectancy or less than a 25% chance of 'survival with benefit' at two years, with benefit implying symptom reduction or improved quality of life [16,17]. The European Society of Cardiology (ESC) broadens this to include lack of efficacy or meaningful survival based on the patient's values [3]. Recent studies echo Arnold's definition, while Puri et al. suggest considering functional, morbidity, and mortality benefits post-TAVI [18]. Lantelme et al. view futility through a prism of ethics, economics, and technology, agreeing on a one-year mortality benchmark [19]. Mack et al. argue against treating futile patients due to lack of survival or increased disability and reduced quality of life post-treatment [20]. Sevilla et al. introduce the concept of 'active aging' as a criterion for TAVI [21], and Gupta et al. emphasize the importance of predicting mortality for informed decision-making [22]. For Higuchi et al., futility is about not achieving expected outcomes, with early patient death considered the ultimate futility [23]. The literature largely agrees on using one-year mortality as a primary indicator of futility [18,23–36]. Symptom improvement, gauged by the NYHA scale, is a secondary consideration, directly affecting the patient's quality of life [24,28,35–38]. The Kansas City Cardiomyopathy Questionnaire is one tool used to assess quality of life changes post-TAVI [37] but others exist. Cardiac events, including heart failure hospitalizations and Major Adverse Cardiac Events, are also relevant [23,34,35].

While many studies quantify futility, the alignment with patient preferences, assessed through Shared Decision Making, is less frequently addressed yet vital to avoid futile treatments [32]. Some argue for considering the patient's ability to perform specific activities and maintain independence as part of the decision-making process [39].

In the context of health economics, TAVI's cost-effectiveness compared to SAVR and medical management is highlighted, with financial considerations becoming increasingly important in treatment selection[16,18,19,21,27,30,33,40–43] and avoiding procedural discomfort and complications is essential if TAVI objectives are unmet [19].

Patient wishes occasionally influence the Heart Team's decisions, sometimes overriding conventional futility assessments [44,45]. Van Mourik et al. advocate for less stringent TAVI futility criteria to include patient-perceived outcomes [24]. International guidelines promote Shared Decision Making as an essential evidence-based approach for integrating patient preferences into medical decisions [3,17,46].

## Cardiac predictors

Heart and Aorta structure can be predictors of poor outcome. In 2023, Zhou et al. find that patients can be classified in 5 different groups of "cardiac damage" before TAVI, and that the classification help predicting higher 2-year all-cause, cardiac death and cardiac rehospitalization [47]. Aorta calcification, might be predictor of cardiac and all-cause mortality after TAVI [48].

The efficacy of TAVI in Low Flow Low Gradient (LFLG) aortic stenosis patients is debatable, particularly when concurrent comorbidities exist. Rodriguez-Gabella *et al.* determined TAVI to be futile—defined as death or no functional improvement within 6 months—in 25% of LFLG patients, a group that also had a higher persistence in NYHA class III or IV at 20% [49]. Concurrent conditions like chronic obstructive pulmonary disease (COPD), atrial fibrillation, or a lower Stroke Volume Index may significantly raise the risk of TAVI futility [50]. Okuno *et al.* found that while LFLG-AS patients with preserved Ejection Fraction (pEF) showed survival rates comparable to High Gradient Aortic Stenosis (HG-AS) patients with pEF, they had poorer functional status at 1-year, and an aortic valve area under 0.8 cm$^2$ was independently linked to futility defined as death or NYHA III or IV status [28].

Mitral Regurgitation (MR) severity has been linked to 2-year mortality (30), and the presence of MR or Pulmonary valve Regurgitation (PR) is associated with 1-year mortality [23]. The correlation between MR, PR, and avoidance of futility has been reinforced by multiple studies [34,51,52]. Kjonas *et al.* also identify previous myocardial infarction as an independent predictor of 1-year mortality [26]. Similarly, a history of heart failure and an ejection fraction below 30% are noted as independent mortality predictors [53].

Farshadmand *et al.* reveal a strong correlation between Cardiac Power Index (CPI) and 1-year mortality, with a critical cut-off value [54]. Agasthi *et al.* highlight CPI as the most impactful variable in predicting 1-year mortality through machine learning analysis, significantly more so than any other variable [55]. Aquino *et al.* suggest Left Atrial Ejection Fraction (LAEF) as an independent mortality predictor, with volumetric measures of the left atrium improving the predictive accuracy of the STS Index from 0.636 to 0.700 [56].

Despite a post-TAVI decrease in valvulo-arterial impedance (Zva), high pre-procedure Zva predicts poorer quality of life and exercise performance [57]. Right Ventricular Dysfunction, as indicated by RVEF < 50%, has been linked to higher mortality and heart failure hospitalizations both univariately and in multivariate models inclusive of clinical risk factors and echocardiographic findings [58]. Also, the right ventricular-pulmonary arterial coupling, associated with preoperative risk stratification can help for predicting 2-years mortality[59].

Elevated systolic pulmonary artery pressure (PASP) over 60mmHg correlates with increased 1-year mortality [26], while PASP levels ≥50 mm Hg are independently associated with all-cause mortality, albeit not with heart failure hospitalizations [60].

Atrial fibrillation, especially in the presence of conditions like chronic kidney disease, is associated with poor outcomes, including all-cause mortality or major adverse cardiac events [50,53]. Some studies noted that NT-proBNP levels can serve as markers of symptom relief post-TAVI, with extreme levels correlating with poor functional outcomes [36,61]. While elevated levels of CA125 [62] and NT-proBNP may predict adverse outcomes post-TAVI, the combination of these biomarkers suggests that CA125 has superior predictive ability [63]. Rodrigues *et al.*, however, caution that biomarkers tend to assist only post-diagnosis and are less effective at predicting TAVI futility [64].

## Non cardiac predictors

In 2020, Tang *et al.* observed that patients with severe comorbidities had significantly lower three-year survival rates free from all-cause mortality and composite cardiovascular outcomes

compared to patients with non-severe comorbidities, demonstrating a multiplicative increase in mortality risk with each additional comorbidity. However, an improvement in quality of life was noted in both groups [65]. Lantelme *et al.* in 2022 underscored the importance of assessing comorbidity burden to identify patients at risk of futile TAVI outcomes. Their approach used a combination of prognostic scales—CAPRI, Charlson, daily activities scores, STS score, and Logistic EuroSCORE—to categorize patients' prognoses and found that those with poor prognosis did not see significant mortality benefits [66]. Earlier, Hermiller *et al.* had incorporated the Charlson Comorbidity Index to predict one-year mortality [67]. Liao *et al.* systematic review revealed that COPD, commonly found in TAVI patients, correlates with increased short-term and long-term mortality. Alongside COPD, BMI, CKD, and the 6 Minute Walk Test were predictors of TAVI futility [68]. Further, studies have consistently shown COPD as a predictor of mortality at two-year and one-year intervals [23,26,34]. The use of home oxygen, though not exclusive to COPD patients, was also predictive of mortality in some studies [27,67].

Maznyscka *et al.* identified higher Plasma Volume Status (PVS) values as indicators of greater mortality and longer hospital stays post-TAVI [69]. PVS at admission was linked to various postprocedural risks and lower survival rates [70]. Finaly, in 2024, Papzoglou *et al.* suggested in a meta-analysis that PVS help refine risk stratification in TAVI [71]. The neutrophil-to-lymphocyte ratio and albumin levels were other noted predictors of adverse outcomes [40,67,72,73]. Hiesh *et al.* analysis across numerous studies determined that lower preoperative serum albumin levels are significant mortality predictors, but rapid post-TAVI improvements in albumin levels could negate poor outcomes, suggesting its limited utility in futility stratification [73,74].

Frailty can be measured by multiple way [75,76] and assessment tools for TAVI patients lack consensus, yet frailty remains a crucial factor associated with poor outcomes [20,30,67,76–80]. Various scales and measurements have been proposed to define frailty, with each demonstrating a relationship to short- and long-term mortality. The Essential Frailty Toolset was found to be the most reliable predictor of one-year mortality or disability [81]. Wheelchair dependency and low body mass index were also associated with one-year mortality, although such findings were based on limited patient numbers [26,27]. Some authors, argue for considering patients with frail status, as frailty can improve post-TAVI [82,83]. However, the exclusion of frail patients due to comorbidities from studies or TAVI treatment can introduce selection bias [26].

Finally, while age is a mortality predictor, it is more frequently utilized within predictive models rather than as a standalone factor [67,84]. The Italian TAVI registry noted an association between male gender and one-year mortality [53].

## Non-predictive variables

In the realm of predictors for TAVI outcomes, certain variables have been investigated but not all hold predictive value. Recently, and despite common belief several studies suggest that the platelet-to-lymphocyte ratio do not predict poor outcome after TAVI [85,86]. Kreidieh *et al.* found that hospitalization within 30 days before TAVI had no significant impact on post-procedural outcomes [87]. Lak *et al.* studied patients with and without cirrhosis and observed no significant differences in one-year mortality rates, new pacemaker implantation rates post-TAVI, readmission rates for heart failure, or major adverse cardiac and cerebrovascular events between the two groups [88]. When examining specific diagnostic tests, the gradient-adjusted Cardiac Power Index (CPI) did not demonstrate better predictive capacity for one-year mortality post-TAVI than the CPI alone [89]. Similarly, isolated spirometry abnormalities were

deemed unreliable; correct classification of COPD did not predict significant postoperative outcomes in TAVI patients [90]. The prognosis for TAVI patients with paradoxical low-flow, low-gradient aortic stenosis (pLFLG-AS) is contentious. Mosleh et al. reported that pLFLG-AS did not correlate with adverse outcomes, noting that TAVI facilitated left ventricular remodeling and comparable mortality rates and functional improvements at one year [37]. Fauchier *et al.* observed that baseline Mitral Regurgitation (MR) was linked to increased cardiovascular and total mortality post-TAVI but did not serve as an independent predictor, suggesting MR alone should not determine futility [91]. Similarly, Amat *et al.* concluded that MR by itself is not a predictive factor [52]. Despite the presence of concomitant cardiac amyloid pathology in one out of eight TAVI patients, it did not affect post-TAVI mortality [92]. Cannatta *et al.* systematic review even suggests that TAVI could be beneficial for patients traditionally considered to be receiving futile care [93].

## Stratification tools

Various medical models aim to predict short-term outcomes like 30-day mortality and longer-term outcomes such as 1-year all-causes mortality [67]. However, models derived from surgical scores generally perform poorly in prediction accuracy [94]. EuroSCORE 2, for example, has been identified by Xuan *et al.* as a weak predictor of short-term survival after TAVI [95]. A 2017 meta-analysis by Wang *et al.* presented c-statistics of only 0.62 for three tests, indicating limited predictive effectiveness [96]. In contrast, Okuno *et al.* suggest the STS Prom as a potentially useful tool for predicting futility in patients with low-flow, low-gradient aortic stenosis (LFLG AS) [28]. Puri et al. summarize findings from three major studies that developed stratification tools from large databases: Partner [97], France2 [98], and TARIS [99]. Their analysis indicates that while these tools outperform the STS and EuroSCORE 2, they only offer modest predictive power, possibly due to not incorporating frailty measures. Their validation scores are 0.64, 0.59, and 0.60, respectively. They recommend a classification for TAVI patients into four risk categories—low, intermediate, high, and prohibitive—based on a composite assessment of risk calculators, the KATZ index for frailty, and failing organ systems [18]. The France2 score, encompassing nine pre-procedural factors, was initially validated for early post-TAVI mortality. Xuan et al. found that a high France2 score (>5) correlated with poor 1-year survival [95]. The Observant score by Capodanno *et al.* demonstrated good discrimination with c-statistics of 0.73 in development and 0.71 in validation, showing better performance than the logistic EuroSCORE 2 [100]. Zusman *et al.* developed the TAVI Futility Risk Model, which showed moderate predictive ability and effectively categorized patients into risk groups based on their score [35]. Fauchier *et al.*'s Futile TAVI Simple score (FTS) also displayed good predictive performance, surpassing the EuroSCORE 2, Charlson comorbidity index, and frailty index in identifying futility [99]. Higuchi *et al.* formulated a model with a c-statistic of 0.73, classifying patients based on the presence of heart failure, moderate to severe mitral or pulmonary regurgitation, and COPD [23]. Martin *et al.*'s integration of frailty components into the UK National TAVI Registry model significantly improved its predictive accuracy for poor outcomes [79]. Similarly, the Essential Frailty Toolset (EFT) enhanced predictions when added to the STS PROM, effectively forecasting 1-year mortality and 30-day death [81]. Machine learning (ML) is making inroads in predictive medicine [101–103]. Alhwiti *et al.* demonstrated that using pre-procedural data, ML models showed promising prediction rates for in-hospital mortality, requiring only a subset of variables for high accuracy [84]. Dagmar *et al.* developed a score with acute kidney injury as a primary variable, outperforming all other models in predicting in-hospital and 1-year mortality [104]. All TAVI Mortality Risk Scores are presented in "Table 2".

**Table 2. Characterization of currently available TAVI mortality risk scores.**

| Study (Year) | Population | Primary Outcome | Variables | AUC |
|---|---|---|---|---|
| Afilalo et al. (2017) [81] | Total: N = 1,020 | 1-year all-cause mortality | 7 different Frailty scale add to STS-prom | 0.784 at best (EFT Scale) |
| Martin et al. (2018) [79] | Total: N = 2624 | 30-day all-cause mortality | three frailty measures add to France2 Observant ACC | 0.62>0.68 0.56>0.64 0.63>0.68 |
| Xuan et al. (2019) [95] | Total: N = 187 | 30-day all-cause mortality 1-year all-cause mortality | France2 risk score (21 point in total) | 0.793 0.679 |
| Capodanno et al. (2014) [100] | Total: N = 1,256 | 30-day all-cause mortality | glomerular filtration rate <45 ml/min (6 points), critical preoperative state (5 points), NYHA class IV (4 points), pulmonary hypertension [4 points], diabetes mellitus (4 points), previous balloon aortic valvuloplasty (3 points), and left ventricular ejection fraction <40% (3 points). | 0.710 |
| Zusman et al. (2018) [35] | Total: N = 2464 Derivation cohort : N = 1532 Validation cohort: N = 656 | 1-year composite of mortality, stroke, and no improvement in NYHA class (vs. baseline) | included diabetes, baseline New York Heart Association functional class, diastolic dysfunction, need for diuretics, mean gradient, hemoglobin level, and creatinine level | 0.690 |
| Fauchier et al. (2019) [91] | Total: N = 47,872 | 1-year all-cause mortality | older age, male sex, history of hospital stay with heart failure, history of acute pulmonary oedema, atrial fibrillation, previous stroke, vascular disease, diabetes, renal disease, liver disease, pulmonary disease, anemia, history of cancer, metastasis and denutrition. | 0.677 |
| Higuchi et al. (2020) [23] | Total: N = 464 | 1-year death and/or hospitalization for heart failure | hospitalization for heart failure, COPD, moderate/severe mitral or tricuspid regurgitation | 0.73 |

While the Heart Team plays a pivotal role in patient stratification [105] by combining measurable and subjective variables, there is no study that quantitatively evaluates their decision-making performance [30]. Catalano *et al.* found statistical models like the STS-PROM to surpass physician estimates made by the Heart Team [106]. However, the ESC and the ACC still consider the Heart Team indispensable for averting futile TAVI procedures, especially given the complex comorbidity profiles in elderly patients [3,17]. Shared Decision Making (SDM) is deemed essential, such as in Canada where a nurse TAVI specialist facilitates patient-Heart Team communication [107]. No studies have systematically evaluated the efficacy of the Heart Team and SDM in reducing futility, but a retrospective comparison by Salihu et al. of Heart Team decisions against Chat-GPT4's treatment recommendations for severe AS revealed a 90% agreement on TAVI choices and 65% for medical treatment or SAVR. This may indicate that the Heart Team considers qualitative factors beyond the scope of ML systems [108].

## Synthesis of results

First, we find multiple concepts in definition of futility in TAVI. Secondly, we find 3 categories of predictive criteria (cardiac, non-cardiac and tested criteria who were not predictive). Finally, different type of medical predictive models. All summarizes in "Table 3".

## Discussion

Our analysis of multiple cohort studies suggests that not all TAVI patients derive benefit from the procedure, with a subset experiencing poor outcomes. A unified definition of futility in TAVI remains elusive, though a one-year survival metric is commonly employed as a benchmark. Nonetheless, a rationale for this specific cut-off is often absent from the discourse. The

**Table 3. Key clinical variables.**

| Futility | | |
|---|---|---|
| **Definition** | | |
| Futility is the failure to achieve the projected outcome, such as improvement in the quality of life, symptoms burden, or life expectancy. | | |
| **Objective Criteria** | | **Subjective Criteria** |
| Mortality <1y | | Patients' goals, Shared Decision |
| Mortality <2y but without "benefit" | | Financial and ethical considerations |
| • Functional benefit | | |
| • Morbidity benefit | | |
| • Symptoms benefit | | |
| **Predictive Criteria** | | |
| **Cardiac** | **Non-Cardiac** | **Fail to predict** |
| Previous Aorta/Cardiac damage | Lung disease | Previous hospitalizations |
| Low Flow Low Gradient (LFLG) | Plasma volume Serum | Cardiac Amyloid Pathology |
| Valvulo arterial impedance | Serum albumin level | Mitral regurgitation |
| Valve disease other the aortic | Neutrophil to lymphocyte ratio | Gradient adjusted cardiac power vs. cardiac power |
| Myocardial Function | Frailty | Spirometry |
| Volumes and ejection fraction of the left atrium | Kidney disease | Cirrhotic patient |
| Right Heart | Demographics data | pLFLG-AS |
| Rhythms disorders | Comorbidities | Platelet-to-lymphocyte ratio |
| Biomarkers | | |
| **Medical predictive models** | | |
| Surgical model | | Clinical predictive model |
| Machine learning | | Heart Team |

benefits of intervention—symptom and functional improvement, and enhanced quality of life—are cited, yet there's notable variation across studies in how these benefits are evaluated.

A multitude of variables, both cardiac and non-cardiac, have been scrutinized to refine the prediction of one-year mortality. While some variables stand out as independent predictors in multiple studies, others, like COPD and the assessment of frailty, continue to provoke debate. Conditions such as low-flow, low-gradient aortic stenosis or cardiac amyloidosis complicate treatment strategies and risk stratification. Alone, these variables can't serve as a stratification tool; they only signal an increased risk without ensuring its manifestation. Thus, they augment the physician's stratification process rather than automate it. To address this complexity, various models have been proposed that integrate multiple variables—sometimes including frailty. These models are heterogeneous and have not shown high reliability. In contrast, machine learning models, utilizing extensive datasets, have yielded statistically significant results. Yet, the practicality of deploying these models in real-life scenarios is a work in progress. Ultimately, the Heart Team carries the responsibility of considering non-quantifiable factors such as patient preferences, financial constraints, and ethical considerations, positioning them at the core of stratification to prevent futile interventions. The most common interpretation of futility involves poor outcomes, typically characterized as patient mortality within one-year post-procedure. The inclusion of quality of life assessments in multiple studies [37,38,57,100] underscores its significance in determining the value of the procedure, linking it to symptom burden and morbidity. The investigation into various predictive variables underscores that no single factor alone suffices for patient stratification. Controversial variables exist, and some have been deemed non-predictive [90]. Specific patient categories, such as those with LFLG AS, necessitate distinct studies for accurate stratification.

Medical prediction models have been rigorously tested across different countries. Despite their sophistication and the inclusion of frailty, these models have not consistently achieved an AUC above 0.8. Compared to established surgical models, the added value of these newer models must be weighed against the resources required for their implementation. Machine learning models appear most promising due to their ability to incorporate numerous variables that older models couldn't manage efficiently [84,104]. However, the Heart Team remains vital for addressing non-quantifiable variables[108].

Advances in technology and patient management may prompt reevaluation of what's considered futile, especially in conditions like AS with concurrent cardiac amyloidosis [93]. Health policy evolution, medical ethics, and international disparities will probably continue to shape the concept of futility.

Furthermore, the concept of futility must be carefully navigated. The goal is to avoid unnecessary procedures without unjustly denying potentially beneficial treatments, as some patient cases have defied the standard assumptions of futility [44,45].

## Limitations

This review's primary limitation is the absence of a consensus definition for futility, which is often equated to "poor outcome" and varies by author. This ambiguity sometimes required assumptions about the author's intent when discussing futility. Moreover, while many studies have investigated predictors of one-year mortality, they did not explicitly address futility. Consequently, these studies were excluded from this review focused on futility. Similarly, discussions on frailty were only included when directly tied to futility, despite the wealth of available research on the topic. Lastly, cross-country ethical considerations, especially regarding resource allocation for TAVI, presented challenges in distilling clear evidence.

Conducting a scoping review was a challenging process. One of the methodological limitations of this article is that it represents the authors' first attempt at performing a scoping review. Despite our rigorous efforts to adhere to international guidelines, we acknowledge that certain improvements could be made with a more experienced perspective.

## Conclusions

Futility is an ethical concept. In TAVI it seemed to be mostly focused on 1-year survival, and modestly on symptoms improvement. There are many variables, cardiac or not, that are found to be independent predictors for poor outcomes. Some variables need more exploration to be unanimous (i.e. LFLG patients, MR). There appears to be a growing consensus that it is important to integrate some measurement for frailty in risk stratification but not how to assess it. It remains challenging to create an efficient, easy to use, medical predicting model. In this domain ML showed promising results with quantitative variables but its potential needs to be demonstrated in more studies to have an impact into practice[103]. On the other hand, the importance of a shared decision model, especially in relation to the Heart Team, appears to be fundamental to incorporate non-quantifiable variables, like patient preferences and goals. We draw two main conclusions. First, a consensus around the definition for futility in relation to TAVI should be developed. Second, future studies should examine the efficacy of the Heart Team and SDM. This latter point seems particularly important in light of the inclusion of SDM in international guidelines, but lack of empirical evidence of their efficacy in prevent futile interventions.

## Supporting information

**S1 Table. PICO framework.**
(PDF)

**S2 Table. Concepts and medical subject headings (Mesh).**
(PDF)

**S3 Table. Search equations.**
(PDF)

**S4 Table. Prisma flow chart (February 2024).**
(PDF)

**S5 Table. 2$^{nd}$ search (08.24) on PubMeb with wider equation.**
(PDF)

**S6 Table. Categories used for data extraction.**
(PDF)

**S7 Table. Included studies related to research questions.**
(PDF)

**S8 Table. Preferred Reporting Items for Systematic reviews and Meta-Analyses extension for Scoping Reviews (PRISMA-ScR) checklist.**
(PDF)

## Acknowledgments

The authors would like to extend their gratitude to Dr. Christopher Rickard for his contributions. Our thanks also go to Professor Petra Schafer and Dr. Gabrielle Dos Santos for their assistance in structuring the methodology of this study. And finally, to Miss Largo Robertini Natacha, librarian, for her guidance in the search process.

## Author Contributions

**Conceptualization:** Charlie Ferry, Stephane Cook.

**Data curation:** Charlie Ferry, Jade Fiery-Fraillon.

**Formal analysis:** Charlie Ferry, Jade Fiery-Fraillon, Mario Togni, Stephane Cook.

**Funding acquisition:** Stephane Cook.

**Investigation:** Charlie Ferry, Jade Fiery-Fraillon.

**Methodology:** Charlie Ferry, Jade Fiery-Fraillon, Mario Togni, Stephane Cook.

**Project administration:** Stephane Cook.

**Software:** Charlie Ferry.

**Supervision:** Mario Togni, Stephane Cook.

**Validation:** Charlie Ferry, Jade Fiery-Fraillon, Mario Togni, Stephane Cook.

**Visualization:** Jade Fiery-Fraillon, Stephane Cook.

**Writing – original draft:** Charlie Ferry.

**Writing – review & editing:** Mario Togni, Stephane Cook.

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
