## [Decision Letter · Decision Letter 0]

14 Aug 2024

PONE-D-24-18956Futility in TAVI: A Scoping Review of Definitions, Predictive Criteria, and Medical Predictive ModelsPLOS ONE

Dear Dr. Ferry,

Thank you for submitting your manuscript to PLOS ONE. After careful consideration, we feel that it has merit but does not fully meet PLOS ONE’s publication criteria as it currently stands. Therefore, we invite you to submit a revised version of the manuscript that addresses the points raised during the review process.

We look forward to receiving your revised manuscript.

Kind regards,

Marina De Rui, MD PhD

Academic Editor

PLOS ONE

Journal Requirements:

2. We noted in your submission details that a portion of your manuscript may have been presented or published elsewhere. [Since this work is a scoping review, the vast majority of the results come from the literature. All literature and authors have been cited.] Please clarify whether this [conference proceeding or publication] was peer-reviewed and formally published. If this work was previously peer-reviewed and published, in the cover letter please provide the reason that this work does not constitute dual publication and should be included in the current manuscript.

4. We note you have included a table to which you do not refer in the text of your manuscript. Please ensure that you refer to Table 3 in your text; if accepted, production will need this reference to link the reader to the Table.

Reviewers' comments:

Reviewer's Responses to Questions

**Comments to the Author**

1. Is the manuscript technically sound, and do the data support the conclusions?

Reviewer #1: Yes

Reviewer #2: Yes

2. Has the statistical analysis been performed appropriately and rigorously? 

Reviewer #1: Yes

Reviewer #2: N/A

3. Have the authors made all data underlying the findings in their manuscript fully available?

Reviewer #1: Yes

Reviewer #2: Yes

4. Is the manuscript presented in an intelligible fashion and written in standard English?

Reviewer #1: Yes

Reviewer #2: Yes

5. Review Comments to the Author

Reviewer #1: Dear Mr. Ferry and colleagues,

Thank you very much for submitting your manuscript to PLOS ONE. This is a very interesting review article that addresses an important topic: futility in TAVI procedures! The choice to produce a scoping review is a good one because this is a very complex subject, and it had to be expected that there will be numerous definitions of futility and especially publications investigating different outcomes’ dependence on it. Therefore, I believe you produced a quite comprehensive overview of the topic describing and highlighting the most important findings from different studies. For my taste, the text is a little bit too much of an enumeration, but probably that’s due to the type of article. From my perspective, there is not much to change. Maybe just reduce the papers that you describe more in detail and concentrate more on the conclusions drawn from your findings.

Technically, I think everything is made perfectly, following all rules to produce a review or a scoping review, respectively.

Concragulations and thank you again for working on this important topic and your efforts to create this paper!

Reviewer #2: Thank you for the opportunity to review this manuscript! My area of expertise is in review methods generally and search methods specifically, so that's what I will be focusing on for my review.

Lines 70-79:

These seem reasonable but I urge the authors to exercise caution when making recommendations or commenting on clinical application since this is a scoping review and the studies were not critically appraised. We can't speak with any kind of certainty as to findings, we can only be descriptive of what we're seeing in the literature (for the most part I think you have achieved this well, but I have flagged a few areas in the manuscript where language could be softened a bit).

Line 82: Please consult PRISMA 2020 for Abstracts as there are a number of reporting items that could be improved (I know it's technically for systematic reviews but since there isn't currently one for scoping reviews many of the items are still relevant!).

Line 90: Please note that PRISMA is a reporting guideline not a conduct guide - please reword this and consider consulting something like the JBI manual, Arksey & O'Malley, Levac, Munn or one of the other seminal scoping review guidance papers.

Line 133: Please state your rationale for using PRISMA for Systematic reviews instead of PRISMA-ScR.

Lines 134-138: Should this be in the protocol/registration section?

Line 141: Please specify which of the 6 Cochrane Library databases were searched (or all of them), as well as platforms that Embase and Cochrane were searched on (Embase.com or Ovid for Embase, Wiley or Ovid for Cochrane).

Line 144: Did the librarian write the search or did they only provide guidance? If the former, have they been extended the offer of co-authorship if they aren't in the authorship list already, and if the latter, have you asked whether they would like to be acknowledged if they haven't been already?

Line 145: Thank you for including your search strategies! This is excellent reporting best practice. When were these searches run?

The search results seems a bit low to me, considering how many databases you searched. Some thoughts on additional subject headings and keywords that you may not have considered: heart valve prosthesis implantation, heart valve prosthesis, valve bioprosthesis, ((transapical or transventricular or percutaneous or transcatheter* or transfemoral or transaxillary) adj3 (valve* or prosthe* or bioprosthe*)), valve replacement, valve implantation; as well as ethics, medical ethics, etc. I also wonder if including risk stratification in your PubMed search is unnecessarily reducing your results. Finally, depending on what platform you searched it on, your Embase search syntax looks incorrect (there should be field codes for the keywords and the /exp is in the wrong location for an Ovid search).

Line 146: Please describe your cross-referencing methods in a bit more detail.

Line 168: Typically scoping reviews will provide a tabular summary of their results (often referred to as "Table 1") where they try their best to summarize the salient information from the various studies. You kind of do this with Table 3, and Appendix 6 but neither provide a broad overview of the literature in tabular form (e.g. citation, population, study or publication type, definition of futility used, predictors reported, tools reported). It might be worthwhile to look at other scoping review examples to see how others have attempted to represent this information (it's definitely challenging, especially with larger included study counts), as this specific table is often the strongest contribution that scoping reviews bring to the literature, as it allows readers to get a high-level overview of the literature study by study very quickly.

Lines 176-208: Great summary!

Lines 209-282: Consider softening some of the language in this section, since none of these studies were critically appraised. For example "Concurrent conditions like COPD, atrial fibrillation, or a lower Stroke volume Index *were reported* to significantly raise the risk of TAVI futility". Reviewing the manuscript with a critical eye for this overall may be helpful.

Lines 402-409: Please also include limitations with regards to the conduct/execution of this study that you are submitting.

Final thoughts: This is generally well conducted and has a few minor clarifications and revisions needed. The biggest weaknesses in my mind at this time are the potentially overly simplistic and/or restrictive searches and the lack of a summary table of results.

6. PLOS authors have the option to publish the peer review history of their article (what does this mean?). If published, this will include your full peer review and any attached files.

Reviewer #1: No

Reviewer #2: No

---

## [Author Response · Author response to Decision Letter 0]

5 Sep 2024

Responses to the reviewers

Dear Dr De Rui,

Dear reviewers, 

Thank you for your thorough review of our manuscript titled Futility in TAVI: A Scoping Review of Definitions, Predictive Criteria, and Medical Predictive Models. 

We appreciate the insightful comments and suggestions provided by the reviewers, which have greatly contributed to improving the quality of our work. In the following response letter, we address each of the points raised in detail. We have carefully considered all feedback and made the necessary revisions to the manuscript. Our responses are structured to correspond directly to the reviewers' comments for clarity and ease of reference.

We hope that our revisions meet your expectations and that the changes have enhanced the manuscript.

Thank you once again for the opportunity to submit the revised version, and we appreciate your time and consideration. We look forward to the possibility of seeing our work published in PLOS One.

Sincerely,

Charlie Ferry, PhD

Responses to the queries

We thank the editors and reviewers for the careful and thoughtful analysis of our manuscript and answer the raised issues below:

Reviews Responses and actions

Journals requirements 

1. Please ensure that your manuscript meets PLOS ONE's style requirements, including those for file naming. The first page has been modified to comply with the guidelines.

The title has been updated, and the tables have been moved from the appendix into the manuscript.

2. We noted in your submission details that a portion of your manuscript may have been presented or published elsewhere. 

We apologize for any misunderstanding, which is likely our fault. However, we would like to clarify that no portion of this study has been published or presented elsewhere. The entire manuscript is original and has not been previously published.

Please confirm at this time whether or not your submission contains all raw data required to replicate the results of your study.

 Thank you for your inquiry. We confirm that this submission contains all the necessary data required to replicate the results of our study. As this article is a scoping review, all the data used have been sourced from existing literature, with no new calculations or modifications made. The search equations used to access the relevant articles are provided in Appendices 1, 2, 3, and 5.

4. We note you have included a table to which you do not refer in the text of your manuscript. Please ensure that you refer to Table 3 in your text. 

Thank you. Table 3 is referenced in the text within the chapter on the stratification tool.

Reviewer #1 

“For my taste, the text is a little bit too much of an enumeration, but probably that’s due to the type of article.” 

Thank you for your feedback. We agree that the text might feel somewhat like an enumeration, which was indeed a challenge. However, this style is somewhat inherent to scoping reviews due to the nature of summarizing a wide range of sources. Nevertheless, we have made some adjustments to a few sentences to improve the flow and readability.

“Maybe just reduce the papers that you describe more in detail and concentrate more on the conclusions drawn from your findings”

 Thank you for your insightful feedback. We understand the concern raised regarding the detailed description of the papers. However, the style of a scoping review necessitates an exhaustive overview of all relevant articles. Reducing and concentrating the information would alter the nature of the review, potentially shifting it towards a different type of review. We appreciate your understanding of the specific requirements of this methodology.

Reviewer #2 

Lines 70-79:

These seem reasonable but I urge the authors to exercise caution when making recommendations or commenting on clinical application since this is a scoping review and the studies were not critically appraised. We can't speak with any kind of certainty as to findings, we can only be descriptive of what we're seeing in the literature (for the most part I think you have achieved this well, but I have flagged a few areas in the manuscript where language could be softened a bit).

 Thank you for your valuable feedback. We have taken your suggestion into account and have softened the language in the results and discussion sections to better reflect the descriptive nature of the findings.

Line 82: Please consult PRISMA 2020 for Abstracts as there are a number of reporting items that could be improved (I know it's technically for systematic reviews but since there isn't currently one for scoping reviews many of the items are still relevant!). 

We acknowledge that eligibility criteria and the name of the information source should align more closely with PRISMA 2020 for Abstracts. However, no formal assessment of bias was conducted, as the review aimed to map the literature. Additionally, we did not use a protocol registration. This is a valuable point, and we will certainly consider it for future reviews.

Line 90: Please note that PRISMA is a reporting guideline not a conduct guide - please reword this and consider consulting something like the JBI manual, Arksey & O'Malley, Levac, Munn or one of the other seminal scoping review guidance papers 

This is an important distinction, thank you.

We rephrase the sentence like this:

“A scoping review was conducted by two researchers and reported in accordance with the PRISMA-ScR guidelines.”

Line 133: Please state your rationale for using PRISMA for Systematic reviews instead of PRISMA-ScR.

 This is a mistake when rewriting the manuscript. We used Prisma for Scoping reviews following the steps on the website.

Lines 134-138: Should this be in the protocol/registration section?

 You're correct. We have renamed this section to 'Protocol' accordingly.

Line 141: Please specify which of the 6 Cochrane Library databases were searched (or all of them), as well as platforms that Embase and Cochrane were searched on (Embase.com or Ovid for Embase, Wiley or Ovid for Cochrane). 

We searched all Cochrane databases, and Embase was accessed via Embase.com using the PICO tool.

Line 144: Did the librarian write the search or did they only provide guidance? If the former, have they been extended the offer of co-authorship if they aren't in the authorship list already, and if the latter, have you asked whether they would like to be acknowledged if they haven't been already? 

Thank you for your comment. The librarian provided guidance during the initial stages of the search process but did not write the search strategy herself. We agree that it is important to acknowledge her contribution, and we will ensure she is properly credited in the acknowledgments section.

Line 145: Thank you for including your search strategies! This is excellent reporting best practice. When were these searches run? 

 I also wonder if including risk stratification in your PubMed search is unnecessarily reducing your results. Finally, depending on what platform you searched it on, your Embase search syntax looks incorrect (there should be field codes for the keywords and the /exp is in the wrong location for an Ovid search). 

Thank you. We included the date (November 2023). 

Thank you. It's true that some of our equations yielded smaller results. For Embase, we utilized their PICO tool, which we deemed sufficient at the time, though we acknowledge that a more refined equation could have been used.

Line 146: Please describe your cross-referencing methods in a bit more detail. 

We reviewed the references of the articles we read in full and identified additional articles that appeared relevant to our research subject. These cross-referenced articles were then further evaluated for inclusion. This process is now described in the 'Search' section of the manuscript.

Line 168: Typically scoping reviews will provide a tabular summary of their results (often referred to as "Table 1") where they try their best to summarize the salient information from the various studies. You kind of do this with Table 3, and Appendix 6 but neither provide a broad overview of the literature in tabular form (e.g. citation, population, study or publication type, definition of futility used, predictors reported, tools reported). It might be worthwhile to look at other scoping review examples to see how others have attempted to represent this information (it's definitely challenging, especially with larger included study counts), as this specific table is often the strongest contribution that scoping reviews bring to the literature, as it allows readers to get a high-level overview of the literature study by study very quickly 

Thank you for the feedback. Given the large number of articles included, it was indeed challenging to create a readable table that covers all the questions comprehensively. However, we provide a more detailed table that offers a clearer overview than Appendix 6, while keeping Table 3 as it is.

Lines 176-208: Great summary! Thank you. 

Lines 209-282: Consider softening some of the language in this section, since none of these studies were critically appraised. For example "Concurrent conditions like COPD, atrial fibrillation, or a lower Stroke volume Index *were reported* to significantly raise the risk of TAVI futility". Reviewing the manuscript with a critical eye for this overall may be helpful. Thank you for your suggestion. We have revised some of the phrasing in this section to soften the language, except in cases where the results are explicitly stated by the original research teams in their articles. In those instances, we have accurately reported their conclusions as presented. However, we have taken special care to avoid making any definitive statements in the discussion and conclusion sections.

Lines 402-409: Please also include limitations with regards to the conduct/execution of this study that you are submitting. We had “Conducting a scoping review is a challenging process. One of the methodological limitations of this article is that it represents the authors' first attempt at performing a scoping review. Despite our rigorous efforts to adhere to international guidelines, we acknowledge that certain improvements could be made with a more experienced perspective.”

Final thoughts: This is generally well conducted and has a few minor clarifications and revisions needed. The biggest weaknesses in my mind at this time are the potentially overly simplistic and/or restrictive searches and the lack of a summary table of results. 

Thank you again for your thorough review. We agree that our search equations were likely a bit too simplistic, although we believe the results provide a comprehensive overview that effectively addresses the research questions. We have updated the summary table as suggested. While we do have additional results, particularly for questions 2 and 3, including them all would have made the table difficult to read given the large number of articles.

---

## [Decision Letter · Decision Letter 1]

24 Sep 2024

PONE-D-24-18956R1Futility in TAVI: A Scoping Review of Definitions, Predictive Criteria, and Medical Predictive ModelsPLOS ONE

Dear Dr. Ferry,

Thank you for submitting your manuscript to PLOS ONE. After careful consideration, we feel that it has merit but does not fully meet PLOS ONE’s publication criteria as it currently stands. Therefore, we invite you to submit a revised version of the manuscript that addresses the points raised during the review process.

**ACADEMIC EDITOR: ** the revised manuscript satisfied the reviewers. For publication a minor revision is needed.

We look forward to receiving your revised manuscript.

Kind regards,

Marina De Rui, MD PhD

Academic Editor

PLOS ONE

Journal Requirements:

Reviewers' comments:

Reviewer's Responses to Questions

**Comments to the Author**

1. If the authors have adequately addressed your comments raised in a previous round of review and you feel that this manuscript is now acceptable for publication, you may indicate that here to bypass the “Comments to the Author” section, enter your conflict of interest statement in the “Confidential to Editor” section, and submit your "Accept" recommendation.

Reviewer #1: All comments have been addressed

Reviewer #2: (No Response)

2. Is the manuscript technically sound, and do the data support the conclusions?

Reviewer #1: Yes

Reviewer #2: Partly

3. Has the statistical analysis been performed appropriately and rigorously? 

Reviewer #1: N/A

Reviewer #2: N/A

4. Have the authors made all data underlying the findings in their manuscript fully available?

Reviewer #1: Yes

Reviewer #2: Yes

5. Is the manuscript presented in an intelligible fashion and written in standard English?

Reviewer #1: Yes

Reviewer #2: Yes

6. Review Comments to the Author

Reviewer #1: Dear Dr. Ferry and colleagues,

From my perspective, you addressed all comments adequately. Thank you very much for the submission of this manuscript and congratulations on your work, which will certainly have an impact in the field!

Best regards

Reviewer #2: Thank you very much for all of your changes. I appreciate the work you have put into this to date. I also agree with the authors that a scoping review should not try to draw conclusions but instead report on what is present in the literature.

Unfortunately, while some of the content I had asked for is present, it's not very well reported (yet!). I would encourage the team to re-review PRISMA-Abstracts, PRISMA-ScR, and PRISMA-Search for reporting, and make sure that each item (if applicable) is reported in the right place (e.g. information sources in the methods section of the abstract, for example). It may be helpful to review the actual statement papers themselves for examples of how such items should be reported and in what detail.

Table 3 and Appendix 6 are excellent and greatly improved, thank you! In Appendix 6 you could further simplify the table by eliminating the "related to research questions" column as its largely repetitive of the much more interesting broken out sections. Also the sub-column "Definition" in the larger "Definition" column seems unnecessary. I think Table 3 needs a bit more contextualization in the methods section for the categories, as not all of them are immediately obvious to the reader (e.g. in what way do previous hospitalizations have to do with a failure to predict? to they not predict futility? if this is part of the extraction form why are we already drawing conclusions here?).

I also still feel that the addition of the "clinical relevance" component of your PubMed search is at odds with the other database searches and as mentioned before, unnecessarily excludes articles. I would encourage the team to conduct and update and leave off the clinical relevance part of the PubMed search and search it from inception (rather than limiting by date as you may do the other databases) so that all the searches are the same across the databases (TAVI + futility). Your review would be greatly strengthened as a result.

7. PLOS authors have the option to publish the peer review history of their article (what does this mean?). If published, this will include your full peer review and any attached files.

Reviewer #1: **Yes: **Simon H. Sündermann

Reviewer #2: No

---

## [Author Response · Author response to Decision Letter 1]

19 Oct 2024

Responses to the queries

We thank the editors and reviewers for the careful and thoughtful analysis of our manuscript and answer the raised issues below:

Journals requirements 

Please review your reference list to ensure that it is complete and correct. If you have cited papers that have been retracted, please include the rationale for doing so in the manuscript text, or remove these references and replace them with relevant current references. 

All the references have been checked and are available. 

Reviewer #2 

Unfortunately, while some of the content I had asked for is present, it's not very well reported (yet!). I would encourage the team to re-review PRISMA-Abstracts, PRISMA-ScR, and PRISMA-Search for reporting, and make sure that each item (if applicable) is reported in the right place (e.g. information sources in the methods section of the abstract, for example). It may be helpful to review the actual statement papers themselves for examples of how such items should be reported and in what detail.

 Thank you for your valuable feedback. We have revised the abstract to include the information sources, as follows: "We identified 129 studies from five key sources: CINAHL, PUBMED, the Cochrane Library, ClinicalTrials.gov, and EMBASE. The literature search was conducted in two rounds—first in February 2024 and again in October 2024—using no restrictions on the year of publication or the language of the studies. Additional references were included through cross-referencing."

Additionally, we recognize that the summary table of results was not placed correctly in the previous version. It should, in fact, be included as Appendix 7 (previously Appendix 6), which compiles all data extracted from the reviewed articles. The synthesis has now been shifted to the end of the Results section for better clarity and coherence. Moreover, the entire article has been reviewed according to PRISMA checklist criteria to ensure transparency and adherence to guidelines.

Appendix 6 you could further simplify the table by eliminating the "related to research questions" column as its largely repetitive of the much more interesting broken out sections. Also the sub-column "Definition" in the larger "Definition" column seems unnecessary. Thank you, Appendix 7 was simplified with the removed of the section “related to question” and the first column “definition”.

I think Table 3 needs a bit more contextualization in the methods section for the categories, as not all of them are immediately obvious to the reader Thank you, I think you may refer to Table 2. For more contextualization we moved this Table in the Results section and added “First, we find multiple concepts in definition of futility in TAVI. Secondly, we find 3 categories of predictive criteria (cardiac, non-cardiac and tested criteria who were not predictive). Finally, different type of medical predictive models. 

I also still feel that the addition of the "clinical relevance" component of your PubMed search is at odds with the other database searches and as mentioned before, unnecessarily excludes articles. I would encourage the team to conduct and update and leave off the clinical relevance part of the PubMed search and search it from inception (rather than limiting by date as you may do the other databases) so that all the searches are the same across the databases (TAVI + futility). Your review would be greatly strengthened as a result.

 Thank you for your insightful comments. In response to your suggestion, we have updated our PubMed search by removing all terms related to "clinical relevance" in the second round of research. As a result, 263 articles were identified, following the same screening process as before. This led to the inclusion of 15 new articles, which strengthened our review(appendix 6). Specifically, two new items were integrated into the results: "Previous Aorta or Cardiac Damage" as a cardiac predictor and "Platelet-to-Lymphocyte Ratio" as a non-predictive variable. This adjustment has enhanced the consistency across all database searches and improved the robustness of our findings

---

## [Editor Report · Decision Letter 2]

24 Oct 2024

Futility in TAVI: A Scoping Review of Definitions, Predictive Criteria, and Medical Predictive Models

PONE-D-24-18956R2

Dear Dr. Ferry,

We’re pleased to inform you that your manuscript has been judged scientifically suitable for publication and will be formally accepted for publication once it meets all outstanding technical requirements.

Kind regards,

Marina De Rui, MD PhD

Academic Editor

PLOS ONE
---

## [Editor Report · Acceptance letter]

5 Nov 2024

PONE-D-24-18956R2 

PLOS ONE

Dear Dr. Ferry, 

I'm pleased to inform you that your manuscript has been deemed suitable for publication in PLOS ONE. Congratulations! Your manuscript is now being handed over to our production team.

Kind regards, 

on behalf of

Dr. Marina De Rui 

Academic Editor

PLOS ONE